

# When ecological marginality is not geographically peripheral: exploring genetic predictions of the centre-periphery hypothesis in the endemic plant *Lilium pomponium*

Gabriele Casazza[1,*], Carmelo Macrì[2,*], Davide Dagnino[2],
Maria Guerrina[2], Marianick Juin[1], Luigi Minuto[2], John D. Thompson[3],
Alex Baumel[1] and Frédéric Médail[1]

[1] Institut Méditerranéen de Biodiversité et d'Ecologie marine et continentale (IMBE),
Aix Marseille Université, Avignon Université, CNRS, IRD. Technopôle de l'Arbois-
Méditerranée, Aix en Provence, France
[2] Department for the Earth, Environment and Life Sciences (DISTAV), University of Genoa,
Genova, Italy
[3] CEFE, Univ Montpellier, CNRS, EPHE, IRD, Univ Paul Valéry Montpellier 3, Montpellier,
France
* These authors contributed equally to this work.

Corresponding author
Gabriele Casazza,
gabriele.casazzabot@gmail.com

## ABSTRACT

**Background:** Quantifying variation of genetic traits over the geographical range of species is crucial for understanding the factors driving their range dynamics. The center-periphery hypothesis postulates, and many studies support, the idea that genetic diversity decreases and genetic differentiation increases toward the geographical periphery due to population isolation. The effects of environmental marginality on genetic variation has however received much less attention.
**Methods:** We tested the concordance between geographical and environmental gradients and the genetic predictions of center-periphery hypothesis for endemic *Lilium pomponium* in the southern Alps.
**Results:** We found little evidence for concordance between genetic variation and both geographical and environmental gradients. Although the prediction of increased differentiation at range limits is met, genetic diversity does not decrease towards the geographical periphery. Increased differentiation among peripheral populations, that are not ecologically marginal, may be explained by a decrease in habitat availability that reduces population connectivity. In contrast, a decrease of genetic diversity along environmental but not geographical gradients may be due to the presence of low quality habitats in the different parts of the range of a species that reduce effective population size or increase environmental constraints. As a result, environmental factors may affect population dynamics irrespective of distance from the geographical center of the range. In such situations of discordance between geographical and environmental gradients, the predictions of decreasing genetic diversity and increasing differentiation toward the geographical periphery may not be respected.

# INTRODUCTION

Quantifying variation of genetic traits over the geographical range of species is crucial for understanding the factors driving their range dynamics and in particular the ecological and evolutionary processes that act on populations on the periphery of a species range (*Thomas et al., 2001*). As climate change begins to induce diverse effects on plant and animal populations (*Thomas et al., 2004*; *Parmesan, 2006*; *Thompson, 2020*) understanding the processes that drive variation in traits across the species' range is of crucial importance.

A major theory that has been developed to enhance our understanding of traits and genetic variation among-population across the species range is the Centre-Periphery Hypothesis (hereafter CPH). The CPH postulates that species' abundance and performance decrease from the geographical center of the range toward the periphery due to the deterioration of environmental conditions (*Hengeveld & Haeck, 1982*; *Brown, 1984*; *Sagarin & Gaines, 2002*; *Gaston, 2003*; *Pironon et al., 2015*, *2017*). A prediction of the CPH is that population genetic diversity is highest in the geographical center of the range and decreases toward the periphery where higher genetic differentiation is predicted as a result of population isolation (*Eckert, Samis & Lougheed, 2008*; *Pironon et al., 2017*). Geographically peripheral populations are also expected to exhibit lower genetic diversity because in such situation they may occur in environmentally marginal habitats (*Eckert, Samis & Lougheed, 2008*). Indeed, demographic processes due to harsh or atypical environmental conditions at the geographical periphery of a species' range can lead to low effective population size that leads to increased inbreeding and low genetic diversity (*Griffin & Willi, 2014*). Moreover, a reduction in genetic diversity may occur in small peripheral populations that experience strong selection and weak migration (*Schoville et al., 2012*). In fact, the CPH often assumes (but see *Pironon et al., 2017*) that geographically peripheral populations are also environmentallu marginal and postulates that a decrease in genetic diversity and an increase in genetic differentiation towards the periphery is due to poor demography, small population size and low density. Peripheral populations may thus be more prone to extinction (*Brown, Stevens & Kaufman, 1996*; *Gaston, 2003*).

However, a concordance between "geographical periphery" and "environmental marginality" (*Brown, 1984*) might not always be correct (*Soulé, 1973*; *Pironon et al., 2017*). In fact, geographically peripheral populations may occur in conditions similar to those in the center of the range (*Piñeiro et al., 2007*; *Kropf, Comes & Kadereit, 2008*) or in different, but not marginal, environmental conditions (*Papuga et al., 2018*). Moreover, variation in ecological factors may impose harsh environmental marginal conditions in any part of the species' range (*Soulé, 1973*). Peripherally isolated populations may also act as source of adaptative diversity under climate change (*Macdonald et al., 2017*; *El Mousadik & Petit, 1996a*, *1996b*; *Thompson, 2020*).
Finally, when different factors are examined (e.g., demographic rates, population size and populations density), they seldom all follow CPH predictions (*Pironon et al., 2017*). This inconsistence is probably because of the complexity of spatial, ecological, and/or historical factors across a geographical gradient (*Eckert, Samis & Lougheed, 2008*; *Pironon et al., 2015*). As a result, although not all expectations are met, some predictions are often supported and the relative importance of the relationships among these factors may be dramatically different from one species to another, resulting in a lack of support for some CPH predictions, even those concerning reduced genetic variation within-population and enhanced differentiation among-populations that are among the most commonly observed trends (*Lira-Noriega & Manthey, 2014*; *Pironon et al., 2017*; *Kennedy et al., 2020*). Several studies suggest that environmental gradients may be more important than geographical gradients for the expression of trends in genetic diversity (*Pilot et al., 2006*; *Cimmaruta, Bondanelli & Nascetti, 2005*; *Lira-Noriega & Manthey, 2014*). This may be due to the fact that populations are less spatially abundant and smaller in marginal environments than near the environmental optimum. This pattern may be particularly important in shaping the spatial distribution of genetic variation in species growing in areas that have remained stable over time (*Nunes, Mancini & Bugoni, 2017*), for example, in climate refugia (*Hewitt, 1999*; *Gavin et al., 2014*; *Hampe & Petit, 2005*), regardless of their current size or density. Unfortunately, very few studies have been conducted on species historical ranges in relation to climate change (*Pouget et al., 2013*; *Kennedy et al., 2020*).

*Lilium pomponium* L. provides an ideal species to test the CPH predictions for genetic variation along geographical and ecological gradients. This plant species has a geographical distribution that is endemic to the Maritime and Ligurian Alps where it occurs along a wide altitudinal gradient. In the study area, a heterogeneous impact of glaciations and topographic complexity probably minimized population extinction during Quaternary glaciations (*Diadema et al., 2005*; *Casazza et al., 2008*, *2016*). Because of the high topographical complexity of the area and the potentially low impact of the glaciation on the distribution range of the species, we expect a low concordance between the geographical and environmental gradient but a close relationship between the genetic variation and environmental gradient. To test these expectations, we addressed the following questions. First, are geographical and environmental center-periphery gradients correlated? Second, are genetic differentiation and genetic diversity associated with geographical and environmental gradients or is there discordance in relationships with those types of gradient?

## MATERIALS AND METHODS

### Study species, data collection and sampling

*Lilium pomponium* L. (Liliaceae) is a perennial herb (geophyte) endemic to the Maritime and Ligurian Alps—a region that represents a major regional hotspot of endemism and plant diversity in the Mediterranean Basin (*Thompson, 2020*). It grows in rocky grassland and shrubland from 100 to 2,000 m of altitude in an area of roughly 7.000 km$^2$ (*Noble & Diadema, 2011*). The species has a self-incompatible outcrossing breeding

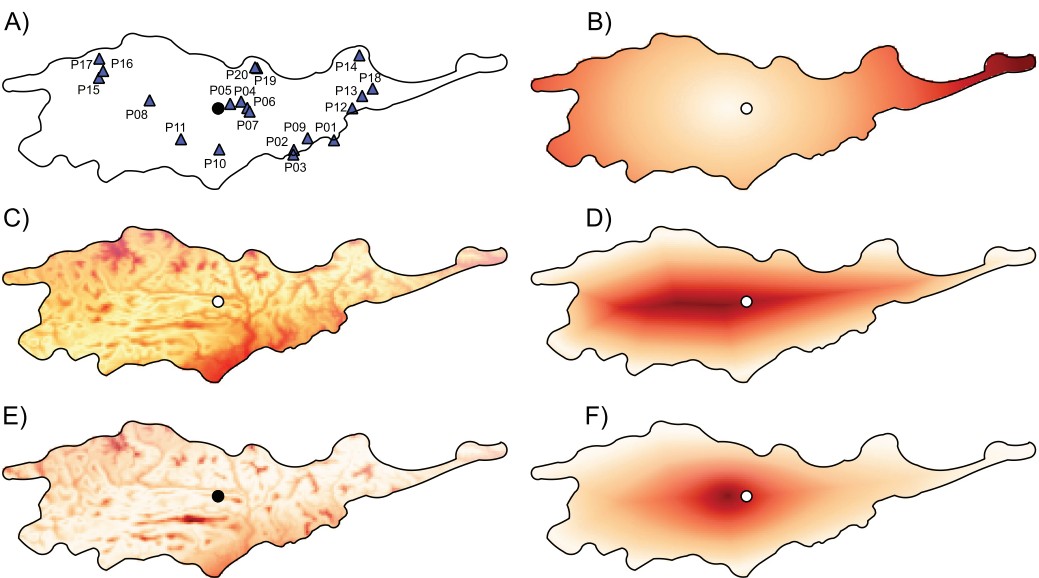

**Figure 1 Geographic location of studied populations of *Lilium pomponium* and the environmental and geographical measures of centrality and, the genetic diversity and differentiation values for each population.** The geographic location of studied populations of *Lilium pomponium* (A) and the environmental (C and E) and geographical (B, D and F) measures of centrality and, the genetic diversity (G) and differentiation (H) values for each population. The spatial distribution of distance from the distributional center (B), the minimum distance from the geographical edge (D) the ratio between edge and center of distribution (F), the Euclidean distance from the centroid of the climatic space (C) and the Mahalanobis distance from the mean of the climatic space (E) is showed. Circle indicates the geographical center and triangles indicate populations sampled. Dark red colors represent high distance and light red colors represent low distance.    

system, with a poor capacity for selfing (*Casazza et al., 2018*). Species occurrences were obtained from field surveys and the Conservatoire Botanique National Méditerranéen SILENE data base (http://www.silene.eu/index.php?cont=accueil) and LiBiOss (Regione Liguria; http://www.cartografiarl.regione.liguria.it/Biodiv/Biodiv.aspx). Overall, a final dataset of 881 occurrence records was used to assess the global distribution of the species and to determine central and peripheral sites and those that may be environmentally marginal. A total of 197 individuals were sampled from 20 populations that cover the entire geographical and altitudinal range of the species (Fig. 1A and Table 1). Field collection was authorized by the Conservatoire Botanique National Méditerranéen de Porquerolles and the Regione Liguria (Decree n. 859 of 22/08/2018).

## DNA extraction and genotyping

Genomic DNA was isolated from fresh leaves according to *Kobayashi et al. (1998)*. Frozen leaves samples were ground in a mixer 150 mill "TissueLyser" (Quiagen-Retsch, Hilden, Germany). Total DNA was extracted using NucleoSpin Plant II 151 Kit (Macherey & Nagel, Germany). DNA concentrations were measured using a photometer (Biophotometer, Eppendorf, Germany). The AFLP method was performed as describe by *Vos et al. (1995)* using laboratory equipment previously employed by *Vidaller (2018)*.
**Table 1 Geographical and environmental distance values for each population.** Geographical and environmental distance values for each population: Population code (code), locality name (Loc.), the distance from the distributional center (dc), the minimum distance from the geographical edge (de), the ratio between edge and center of distribution (dec), the Euclidean distance from the centroid of the climatic space (eu) and the Mahalanobis distance from the mean of the climatic space (ma) are reported.

| code | Loc. | dc (km) | de (km) | dce | eu | ma |
|------|------|---------|---------|-----|-----|-----|
| P01 | Baisse Saint-Paul (FR) | 44.80 | 1.84 | 0.040 | 2.910 | 2.900 |
| P02 | Plateau Tercier (FR) | 35.12 | 2.36 | 0.063 | 2.537 | 2.726 |
| P03 | Fort de la Revère (FR) | 36.13 | 0.64 | 0.017 | 2.809 | 3.704 |
| P04 | Les Pras (FR) | 13.44 | 23.66 | 0.638 | 2.621 | 2.820 |
| P05 | Ciamp du Var (FR) | 9.97 | 24.83 | 0.713 | 2.669 | 2.846 |
| P06 | Tournefort (FR) | 15.14 | 25.11 | 0.624 | 2.473 | 2.198 |
| P07 | Utelle (FR) | 15.90 | 23.14 | 0.593 | 2.325 | 1.718 |
| P08 | Entrevaux (FR) | 15.91 | 25.69 | 0.617 | 1.864 | 2.077 |
| P09 | Gorbio, Col de la Madone (FR) | 36.27 | 5.99 | 0.142 | 2.744 | 5.450 |
| P10 | Col de Vence (FR) | 19.51 | 11.27 | 0.366 | 2.975 | 3.679 |
| P11 | Greolieres (FR) | 14.89 | 17.45 | 0.540 | 1.512 | 0.649 |
| P12 | Mt. Comune (IT) | 47.93 | 12.72 | 0.210 | 0.955 | 0.677 |
| P13 | Mt. Lega (IT) | 51.48 | 15.76 | 0.234 | 0.706 | 0.313 |
| P14 | Castel Tournou (FR) | 55.61 | 0.50 | 0.009 | 1.536 | 0.729 |
| P15 | Méailles (FR) | 34.27 | 9.92 | 0.224 | 1.904 | 1.438 |
| P16 | Peyresq (FR) | 34.23 | 7.99 | 0.189 | 2.397 | 3.502 |
| P17 | Ondres (FR) | 37.66 | 2.69 | 0.067 | 3.627 | 4.140 |
| P18 | Mt. Grai (IT) | 55.24 | 14.98 | 0.213 | 1.977 | 2.527 |
| P19 | L'adrechas (FR) | 25.57 | 8.47 | 0.249 | 2.069 | 1.293 |
| P20 | La Colmiene (FR) | 25.09 | 8.52 | 0.254 | 2.683 | 2.209 |

AFLP genotyping was based on the protocol described by *Vos et al. (1995)*. A total of 100 ng of DNA were digested using the restriction enzymes Eco RI and Tru 9I (Fisher Scientific, France) for 3 h at 37 °C and then for 3h at 65 °C in a total volume of 25 µl (15 µL + 10µl of DNA). Digestion products were ligated to 0.5 µL Eco and 25 µL Mse adaptors for 3 h at 37 °C and treated with T4 DNA Ligase and 0.1 µL of 100 mM ATP to a final volume of 25 µL (5 µL + 20 µl of restriction products). Ligation products were diluted eight times and pre-selective PCR amplification was performed using EcoR1+A, Mse+C primers and *Taq* DNA polymerase in a 44.5 µL volume. The profile of pre-amplification thermocycle was 94 °C for 2 min, followed by 20 cycles at 94 °C for 45 s, 56 °C for 45 s, 72 °C for 1 min and 72 °C for 10 min. For the selective amplification, three primer combinations were chosen for PCR: ASII: EcoR1-AGG/MseI-CGG, ASIII: EcoR1-AGC/MseI-CAG, ASVII: EcoR1-AGC/MseI-CTG dyed with 6-FAM fluorescence at 5′ Eco end (Eurofins Genomics, Ebersberg, Germany).

One Hundred times diluted pre-amplification products were used to perform selective amplification in a final volume of 20 µL (15 µL + 5 µL of diluted pre-amplification products). For the selective amplification thermocycle we used 94 °C for 2 min, 10 cycles of

94 °C for 30 s, 65 °C for 30 s (step −1 °C per cycle), 72 °C for 1 min, followed by 22 cycles at 94 °C for 30 s, 56 °C for 30 s, 72 °C for 1 min, 72 °C for 5 min and 4 °C for 2h. The fragment length produced by the amplification was separated and quantified by electrophoresis using an ABI 3730xl DNA analyzer (Applied Biosystems, Foster City, California, USA) with GS600 LIZ size marker. Peaks were scored in Peak Scanner V 1.0 (Applied Biosystems, Foster City, California, USA) as present (1), absent (0), or "no data" (NA). We utilized Raw Geno 2.0 (*Arrigo et al., 2009*) to select the fragments longer than 100 bp and smaller than 200 bp. Maximum binning between peaks was set at 1.75 and minimum at 1.5. The error rate was calculated as mismatches ratio over matches in 32 replicated individuals (*Bonin et al., 2004*).

## Estimation of geographical and environmental center-periphery gradient

We used different measures to describe the geographical and environmental center-periphery gradients. To identify geographically central and peripheral populations we calculated the geographical center of distribution by averaging the coordinates of the occurrences and the distributional limits by calculating the minimum convex hull polygon that included all the known occurrences of *L. pomponium*. Following this, geographical peripheral populations were identified by calculating for each occurrence its distance from the geographical center, its shortest distance to the edge of the convex hull and the ratio between these two distance measures.

To assess environmental marginality nineteen bioclimatic variables representative of the period 1979–2013 were downloaded from the Chelsa climate database website (*Karger et al., 2017a*, *2017b*; www.chelsa-climate.org) at 30-s (c. 1 km) spatial resolution. We extracted values of bioclimatic variables from cells where the species occurs and defined the climatic space occupied by the species on the basis of the first two axes of a principal component analysis (PCA) using all cell values. Environmental marginality was assessed by calculating the Euclidean distance from the centroid of the climatic space and the Mahalanobis distance from the mean of the climatic space.

## Data analysis

Genetic diversity was measured for each population by calculating Nei's diversity index ($h$) and Shannon's information index ($I$) using POPGENE v. 1.32 (*Yeh, Yang & Boyle, 1999*). Because the various measures of population differentiation quantify different aspects of population structure, we estimated genetic differentiation among populations by using three different measures: the fixation index $F_{ST}$, estimated by using ARLEQUIN 3.11 (*Excoffier & Lischer, 2010*) and Rho (*Ronfort et al., 1998*) and Jost's $D$ (*Jost, 2008*) pairwise distance matrix using GenoDive 3.0 (*Meirmans, 2020*). To estimate the genetic differentiation among-population, we calculated the average of pairwise $F_{ST}$, Rho and $D$ values for each population with all other populations. We calculated Pearson product moment correlations between geographical and environmental center-periphery distance measures and between genetic parameters and the center-periphery distance measures.

# RESULTS

## Geographical and environmental gradients

The first two axes of PCA of climatic data explained 79.71% of the overall climatic variance (29.03% and 50.68%, respectively in Fig. S1). The first PCA axis is positively related to temperature-related parameters and negatively related to precipitation during the warm and dry season. The second axis is positively related to precipitation during the wet and cool seasons and negatively related to temperature seasonality indices. Sites that are marginal in climatic space occurred throughout the entire distributional range, as witnessed by the occurrence of sites that are far from the climatic centroid both near the geographical center and in northern and southern geographically peripheral sites (dark red areas in Figs. 1C and 1E; Table 1).

The distance from the geographical center was higher in eastern and western peripheral sites than in northern and southern peripheral sites (dark red areas in Fig. 1B). As a result, sites farthest from the distributional edges occurred in a long and thin belt close to the center of distributional range (dark red areas in Fig. 1D). Likewise, areas with a low ratio between distance from the edge and the center of distribution occurred mainly at the eastern and western geographical periphery (light red areas in Fig. 1F). In line with these results, the distance from the climatic centroid was not correlated ($r = 0.09$ and $r = 0.05$ according to Euclidean and Mahalanobis distance, respectively) with distance from the geographical center (Figs. 1, 2A and 2D; Table 1) and was only poorly negatively correlated with distance from distributional edge and the edge/centre distance ratio (Figs. 1, 2B, 2C, 2E and 2F; Table 1).

## Genetic diversity

The three AFLP primer pairs produced 134 polymorphic fragments for 197 individuals (Table 2). The AFLP error rate based on 32 replicates was 2.02%. Genetic diversity indices per population ranged from 0.11 to 0.30 with a mean of 0.22 (±0.04) and from 0.19 to 0.43 with a mean of 0.34 (±0.06) according to Nei's diversity index and Shannon information index, respectively (Table 3). Average $F_{ST}$ between each population and all remaining populations ranged from 0.034 to 0.152, the average Jost's $D$ ranged from 0.032 to 0.108 and the average Rho ranged from 0.008 to 0.321 (Table 3). Populations with low and high genetic diversity were scattered throughout the distributional range (Figs. 3A and 3B). In particular, populations with lowest values occurred near the geographical center (i.e., P05), as well as near both the southern (i.e., P01) and the northern (i.e., P20) periphery of the distribution. Similarly, populations with high values occurred both near the geographical center (i.e., P07 and P11) and near the periphery (i.e., P14 and P16).

Populations with low and high $F_{ST}$ and Jost's $D$ values were scattered throughout the distributional range (Figs. 3C and 3D). In contrast, populations with high Rho values occurred mainly near the geographical periphery (i.e., P01, P13, P16 and P17 in Fig. 3E) while populations with low Rho values occurred near the geographical center (i.e., P04, P06 and P07). In line with this pattern, intra-population genetic diversity was not

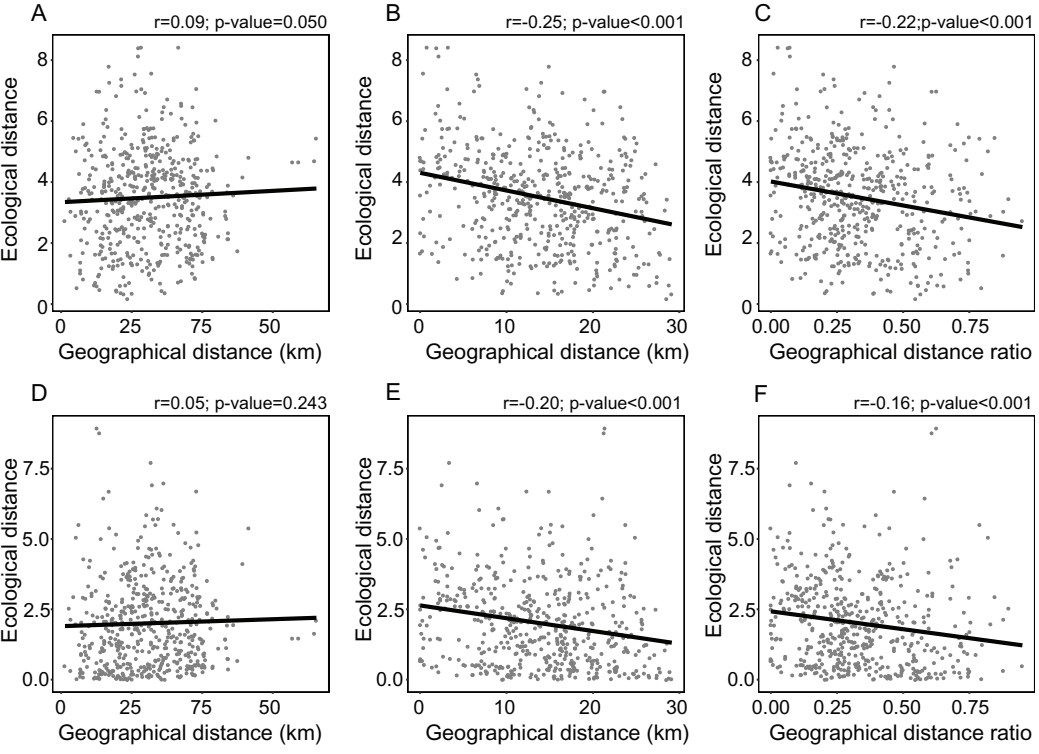

**Figure 2 Linear regressions showing the relationships between measures of environmental marginality and geographic peripherality in *Lilium pomponium*.** Linear regressions showing the relationships between measures of environmental marginality and geographic peripherality in *Lilium pomponium*. The relationship between the Euclidean distance from the centroid of the climatic space and distance from the distribution center (A), the minimum distance from the geographical periphery (B) and the ratio between periphery and center of distribution (C), and the relationship between the Mahalanobis distance from the mean of the climatic space and the distance from the distributional center (D), the minimum distance from the geographical periphery (E) and the ratio between the periphery and center of distribution (F).

significantly correlated with geographical distance measures (Table 4). Two measures of average inter-population differentiation (i.e., $F_{ST}$ and Jost's $D$) were not correlated with measures of geographical marginality while Rho index was significantly correlated with measures of geographical marginality, increasing toward the geographical periphery (Table 5). Genetic diversity indices were significantly correlated with both distances from climatic centroid (Table 4) while none of the genetic differentiation indices per population were significantly correlated with environmental distances (Table 5).

## DISCUSSION

In this study on *Lilium pomponium*, the central-periphery hypothesis (CPH) assumptions of concordance between geographical and environmental gradients are not supported neither is the prediction of decreasing genetic diversity towards the geographical periphery. Our results nevertheless provide support for the prediction of increasing differentiation towards the range periphery.

**Table 2 Primer pairs used in selective amplifications and summary of the number of AFLP fragments scored.**

| *Eco*RI primer | *Mse*1primer | Number of loci |
|---|---|---|
| E-AGC | M-CGG | 78 |
| E-AGC | M-CAG | 47 |
| E-AGC | M-CTG | 85 |
| Total | | 210 |
| Mean | | 70 |

**Table 3 Gene diversity estimates in the total dataset based on AFLP data.** Gene diversity estimates in the total dataset based on AFLP data: Population code (code), locality name (Loc.), estimated population size (Size), sample size ($N$), Nei's diversity index ($h$), Shannon's information index ($I$) and, the average of the pair of $F_{ST}$, Jost's $D$ ($D$) and Rho values between each population and all remaining populations and associated standard deviations (in brackets) are reported.

| code | Loc. | Size | h | I | $F_{ST}$ | D | Rho |
|---|---|---|---|---|---|---|---|
| P01 | Baisse Saint-Paul (FR) | ~100 | 0.1141 | 0.1899 | 0.0874 (0.0310) | 0.0933 (0.0750) | 0.2004 (0.2561) |
| P02 | Plateau Tercier (FR) | ~250 | 0.1930 | 0.3140 | 0.0874 (0.0572) | 0.0479 (0.0544) | 0.0119 (0.1443) |
| P03 | Fort de la Revère (FR) | ~40 | 0.2025 | 0.3194 | 0.0344 (0.0801) | 0.0454 (0.0449) | 0.1054 (0.2352) |
| P04 | Les Pras (FR) | ~50 | 0.2022 | 0.3229 | 0.1408 (0.0591) | 0.0416 (0.0381) | 0.0076 (0.1621) |
| P05 | Ciamp du Var (FR) | ~30 | 0.1582 | 0.2535 | 0.0551 (0.0355) | 0.0711 (0.0671) | 0.0076 (0.1621) |
| P06 | Tournefort (FR) | ~200 | 0.2363 | 0.3651 | 0.0711 (0.0363) | 0.0357 (0.0252) | 0.0526 (0.1059) |
| P07 | Utelle (FR) | ~100 | 0.2489 | 0.3840 | 0.0934 (0.0462) | 0.0351 (0.0200) | 0.0076 (0.1621) |
| P08 | Entrevaux (FR) | ~20 | 0.2508 | 0.3885 | 0.1132 (0.0601) | 0.0318 (0.0206) | 0.0405 (0.2031) |
| P09 | Gorbio, Col de la Madone (FR) | ~200 | 0.2102 | 0.3332 | 0.0642 (0.0450) | 0.0379 (0.0446) | 0.1054 (0.2352) |
| P10 | Col de Vence (FR) | ~200 | 0.2514 | 0.3837 | 0.0691 (0.0358) | 0.0639 (0.0275) | 0.1459 (0.1406) |
| P11 | Greolieres (FR) | ~150 | 0.2585 | 0.3907 | 0.0615 (0.0660) | 0.0484 (0.0256) | 0.0127 (0.1968) |
| P12 | Mt. Comune (IT) | ~150 | 0.2422 | 0.3627 | 0.0942 (0.0744) | 0.1079 (0.0538) | 0.0266 (0.1239) |
| P13 | Mt. Lega (IT) | ~200 | 0.2878 | 0.4346 | 0.1161 (0.0867) | 0.0533 (0.0269) | 0.3209 (0.1862) |
| P14 | Castel Tournou (FR) | ~30 | 0.2466 | 0.3768 | 0.0535 (0.0529) | 0.0393 (0.0196) | 0.0098 (0.1216) |
| P15 | Méailles (FR) | ~250 | 0.2185 | 0.3380 | 0.0740 (0.0574) | 0.0392 (0.0347) | 0.0098 (0.1216) |
| P16 | Peyresq (FR) | ~70 | 0.2293 | 0.3452 | 0.0620 (0.0520) | 0.1054 (0.0438) | 0.3209 (0.1862) |
| P17 | Ondres (FR) | ~200 | 0.1989 | 0.3118 | 0.0896 (0.0410) | 0.0564 (0.0353) | 0.3209 (0.1862) |
| P18 | Mt. Grai (IT) | ~500 | 0.2738 | 0.4115 | 0.0868 (0.0353) | 0.0524 (0.0251) | 0.1054 (0.2352) |
| P19 | L'adrechas (FR) | ~400 | 0.2175 | 0.3326 | 0.1524 (0.0479) | 0.0852 (0.0356) | 0.0405 (0.2031) |
| P20 | La Colmiene (FR) | ~200 | 0.1758 | 0.2806 | 0.1043 (0.0424) | 0.0451 (0.0428) | 0.0394 (0.1854) |

## Lack of concordance among geographical and environmental gradients

An implicit assumption of the CPH is a symmetrical, monotonic deterioration of environmental condition from the center of the distribution toward the periphery (*Brown, 1984*). However, in mountainous environments, such as the Maritime and Ligurian Alps where *L. pomponium* occurs, environmental factors change over very short distances because of high topographic complexity (*Körner, 2003*; *Casazza et al., 2008*; *Thompson, 2020*), imposing environmentally marginal conditions in scattered and diffuse parts of a

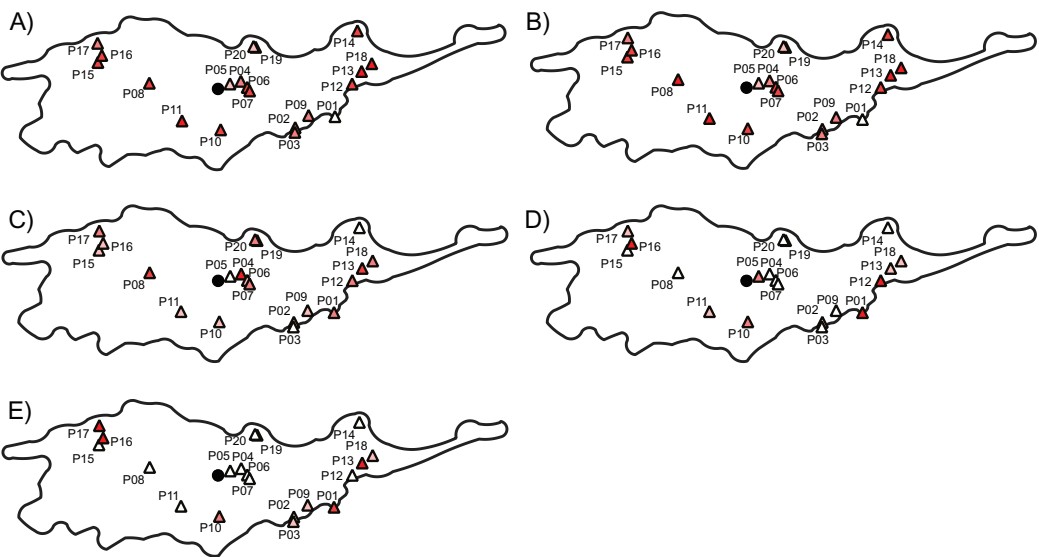

**Figure 3 The genetic diversity and differentiation values for each studied populations of *Lilium pomponium*.** The genetic diversity (A and B) and differentiation (C and D) values for each studied populations of *Lilium pomponium*. Genetic diversity was measured by calculating Nei's diversity index (A) and Shannon's information index (B). Genetic differentiation among populations was estimated by using $F_{ST}$ (C), Jost's $D$ (D) and Rho (E). Circle indicates the geographical center and triangles indicate populations sampled. Dark red colors represent high value and light red colors represent low value. Pearson's product moment correlations and significance values are reported above each plot.

**Table 4 Pearson's correlation (*r*) and the significance (*p*-value) of genetic diversity vs geographical and environmental distances.**

|  |  | h | | I | |
|---|---|---|---|---|---|
|  |  | *r* | *p*-value | *r* | *p*-value |
| Geographical | Centre distance | 0.319 | 0.171 | 0.116 | 0.624 |
|  | Edge distance | 0.142 | 0.551 | 0.328 | 0.158 |
|  | Edge/centre distance | 0.169 | 0.476 | 0.182 | 0.441 |
| Environmental | Euclidean distance | −0.761 | <0.001 | −0.730 | <0.001 |
|  | Mahalanobis distance | −0.513 | 0.020 | −0.479 | 0.032 |

given species' range. Moreover, topographic heterogeneity may decouple site climatic conditions from regional climatic values, reducing our capacity to detect causal factors related to demographic process and environmental variation (*Hannah et al., 2014*; *Patsiou et al., 2014*).

In *L. pomponium* climatically marginal conditions were found both at the periphery of distribution and close to the geographical center, mainly on valley floors (dark red areas in Figs. 1C and 1E; Table 1). This scattered distribution of central and marginal environmental conditions in different parts of the distributional range of this species are probably favored by the succession of high and low elevation areas. In fact, in the study region, tectonic, glacial and fluvial processes have shaped the deeply incised valleys that are

**Table 5 Pearson's correlation ($r$) and the significance ($p$-value) of genetic differentiation vs geographical and environmental distances.**

| | | $F_{ST}$ | | Jost's $D$ | | Rho | |
|---|---|---|---|---|---|---|---|
| | | $r$ | $p$-value | $r$ | $p$-value | $r$ | $p$-value |
| Geographical | Centre distance | −0.115 | 0.630 | 0.248 | 0.292 | 0.397 | 0.083 |
| | Edge distance | 0.276 | 0.238 | −0.242 | 0.305 | −0.324 | 0.164 |
| | Edge/centre distance | 0.207 | 0.381 | −0.255 | 0.277 | −0.407 | 0.075 |
| Environmental | Euclidean distance | 0.015 | 0.949 | −0.064 | 0.787 | −0.011 | 0.964 |
| | Mahalanobis distance | −0.225 | 0.349 | −0.059 | 0.803 | 0.144 | 0.544 |

characteristic of the southern Alps (*Sanchez et al., 2010*). As a result, the lack of correlation between geographical and environmental distances is not surprising (Fig. 2).

In addition, the distribution range of *L. pomponium* is largely west-east orientated because the north-south extension is constrained by the sharp altitudinal gradient of the mountain chain. As a result, the east and west peripheral populations are further away from the center than south and north peripheral populations (Figs. 1B–1F). For this reason, harsh conditions in the latter are closer to the geographical center than harsh conditions in the former peripheral populations (dark red areas in Figs. 1C and 1E). This asymmetry may explain why correlations are higher between environmental gradients and distance from the edge of distribution than between environmental gradients and the distance from the geographical center (Figs. 2B, 2D and 2F). In *L. pomponium*, the weak association between geographical and environmental distances is thus closely affected by the complex topography of the region that causes a discrete and strongly asymmetrical distribution of environmental conditions across the distributional range. Correlations between geographical and ecological gradients thus may not occur in areas of high localized spatial heterogeneity. Indeed, the occurrence of central environmental conditions at the geographical margins (white areas in Figs. 1C and 1E) suggests that in *L. pomponium* range limits may be due to other factors, such as a gradient in habitat availability, rather than a gradient in habitat quality. This supports the idea that local environmental conditions (such as in microrefugia) explain the occurrence of a patchy distribution (*Patsiou et al., 2014*).

## Genetic structure and the CPH

In general, intra-population genetic diversity is expected to be associated with inter-population differentiation because stochastic genetic drift due to change in population size causes both a decline in genetic diversity within populations and divergence among populations (*Eckert, Samis & Lougheed, 2008*). In fact, peripheral populations having a low genetic diversity usually have also a high genetic differentiation (*Pironon et al., 2017*). However, our results do not provide support for the CPH prediction of a reduction in within-population genetic diversity towards the periphery but do provide evidence for an increase in inter-population genetic differentiation in peripheral populations relative to those in the center.

For *L. pomponium*, although Rho measure of genetic differentiation illustrate high genetic differentiation near the geographical periphery, suggesting a decrease in gene flow among peripheral populations, we found no evidence for a decline in genetic diversity in peripheral populations (Fig. 3; Tables 3 and 4). The difference between Rho and both $F_{ST}$ and Jost's *D* is probably because of Rho is independent from the selfing rate, while the other measures increase when self-fertilization occurs within populations (*Ronfort et al., 1998*). This may explain why some small-sized populations near the geographical center but far from environmental center (i.e., P04, P05 and P08 in Table 3) have high $F_{ST}$ and Jost'*D* values and low Rho values. In fact, the high number of flowers per inflorescence detected in these small populations may favor geitonogamous pollination among flowers on the same plant (*Macrì et al., 2021*). In small sized populations of facultative-autogamous species, like *L. pomponium* (*Casazza et al., 2018*), geitonogamy may assure the production of a small, but regular number of seeds (*Roberts et al., 2014*), but at the cost in terms of reduced outcrossing (*Lloyd, 1992*; *Harder & Barrett, 1995*).

The high genetic differentiation among peripheral populations suggested by Rho index may be the result of a decrease in the availability of suitable habitat and therefore a more (naturally) fragmented distribution of populations rather than the result of a decline in habitat quality and population size (*Pironon et al., 2017*). In fact, the reduction in density of suitable habitat may reduce population connectivity both by pollen and seeds (*Hargreaves & Eckert, 2014*; *Young, Boyle & Brown, 1996*), favoring isolation of populations and resulting in higher inter-population differentiation. Moreover, in *L. pomponium* population connectivity may be further reduced by a decrease in suitable habitat due to (i) shrubland or forest expansion in mid-elevation of high elevation sites (*Carlson et al., 2014*) and/or (ii) urbanization in peripheral coastal lowland areas (*Noble & Diadema, 2011*). Given that gene flow in perennial plants mainly occurs by pollen transfer (*Levin & Kerster, 1974*; *Ennos, 1994*; *Tarayre et al., 1997*), pollen limitation detected in the southernmost populations of *L. pomponium* may further drive the genetic differentiation among these populations (*Macrì et al., 2021*).

In general, diversity is expected to decline towards the geographical periphery, regardless of environmental conditions (*Eckert, Samis & Lougheed, 2008*; *Pironon et al., 2017*). However, in *L. pomponium* genetic diversity does not decrease with the distance from the geographical center but does decline with distance from climatic centroid (Table 4). The lack of association between genetic diversity and distance from the geographical periphery is indicative of historical range stability, and of the existence of patchwork of many local refugia, the so-called refugia within refugia hypothesis. The occurrence of widespread local refugia characterizes the Mediterranean region, explaining the persistence of species during the Pleistocene glaciation cycles (*Médail & Diadema, 2009*; *Thompson, 2020*) and their ability to rapidly recolonize new locations at higher altitude or latitude at the end of glaciation (*Villellas et al., 2014*; *Ferreira et al., 2015*).

In species survived in few glacial refugia, patterns of genetic variation have been strongly shaped by past climate-driven range dynamics, and consequently a decrease of within-population genetic diversity is expected towards recently colonized areas at high

altitude or latitude because of the stochasticity of colonization processes (*Hampe & Petit, 2005*; *Thompson, 2020*). However, contrary to this expectation, in *L. pomponium* we detected the highly genetically diverse populations at high latitude and elevation and the less genetically diverse populations at low altitude or latitude edge. This finding supports the hypothesis of a past range expansion and a recent contraction in relation with the complex phylogeographical history of the Maritime and Ligurian Alps (*Médail & Diadema, 2009*). A further analysis aimed precisely at reconstructing past range dynamics would be most interesting here. In contrast to what we have observed in *L. pomponium*, high genetic diversity near the geographical center has been recorded in other Mediterranean plant species in which a central–marginal pattern may be associated with a history of expansion from a center of diversity (*Pouget et al., 2013*). Elsewhere, the pattern we observed in *L. pomponium* has been previously detected in 40 species in which environmental distance more closely predicts genetic diversity than geographical distance (*Lira-Noriega & Manthey, 2014*).

The reduction of neutral genetic diversity in environmentally marginal conditions may be due to a cascade effect of environmental impacts on population dynamics (*Lira-Noriega & Manthey, 2014*; *Pironon et al., 2017*). The quality of the local habitat (i.e., distance from the environmental center) may affect demographic processes (*Jaquiéry et al., 2008*), ultimately resulting in a reduction in genetic variation due to selection and/or genetic drift. Nevertheless, in *L. pomponium* some small populations have relatively high genetic diversity (e.g., P03, P14 and P16 in Table 3). The perennial nature of *L. pomponium* (more than 10 years in *L. martagon*; *Lundqvist, 1991*) may reduce the effect of genetic drift (*Ellstrand & Elam, 1993*; *Nybom & Bartish, 2000*). In addition, natural selection may directly play an important role in shaping neutral genetic diversity (*Gillespie, 2000*; *Leffler et al., 2012*; *Corbett-Detig, Hartl & Sackton, 2015*). In large populations that incur strong environmental filtering, natural selection may directly affect levels of neutral variation and favor variants that are linked to beneficial mutations (*Gillespie, 2000*) and/or purge those linked to deleterious mutations (*Kaplan, Hudson & Langley, 1989*), the so-called genetic draft (*Gillespie, 2000*). In *L. pomponium* genetic differentiation may thus be affected by habitat availability that reduces population connectivity while genetic diversity may be affected by habitat quality that affects population dynamics.

## CONCLUSIONS

In conclusion, our study provides support for CPH predictions of increased differentiation toward the geographical limits of the distribution of this species (*Lira-Noriega & Manthey, 2014*; *Kennedy et al., 2020*). However, we found neither evidence for increased environmental marginality nor a reduction in within-population genetic diversity towards the geographical periphery. Taken together our results suggest that habitat quality may affect genetic diversity both by a cascade effect on population dynamics and by purging neutral variation that is linked to unfavorable mutations. A reduction in available habitat, rather than a decline in environmental quality, may explain the increase in inter-population genetic differentiation toward the geographical periphery of the species' range.

## ACKNOWLEDGEMENTS

The authors thank Katia Diadema and Maëlle Le Berre (Conservatoire botanique national méditerranéen) for their help in populations' sampling.

### Funding

This work was supported by the European Union's Horizon 2020 research and innovation program under grant agreement No. 793226. The funders had no role in study design, data collection and analysis, decision to publish, or preparation of the manuscript.

### Grant Disclosures

The following grant information was disclosed by the authors:
European Union's Horizon: 793226.

### Competing Interests

Gabriele Casazza is an Academic Editor for PeerJ.

### Author Contributions

- Gabriele Casazza conceived and designed the experiments, analyzed the data, prepared figures and/or tables, authored or reviewed drafts of the paper, and approved the final draft.
- Carmelo Macrì conceived and designed the experiments, performed the experiments, analyzed the data, authored or reviewed drafts of the paper, and approved the final draft.
- Davide Dagnino analyzed the data, authored or reviewed drafts of the paper, and approved the final draft.
- Maria Guerrina analyzed the data, authored or reviewed drafts of the paper, and approved the final draft.
- Marianick Juin performed the experiments, authored or reviewed drafts of the paper, and approved the final draft.
- Luigi Minuto conceived and designed the experiments, authored or reviewed drafts of the paper, and approved the final draft.
- John D. Thompson conceived and designed the experiments, authored or reviewed drafts of the paper, and approved the final draft.
- Alex Baumel conceived and designed the experiments, performed the experiments, authored or reviewed drafts of the paper, and approved the final draft.
- Frédéric Médail conceived and designed the experiments, authored or reviewed drafts of the paper, and approved the final draft.

### Field Study Permissions

The following information was supplied relating to field study approvals (i.e., approving body and any reference numbers):

Field collection was approved by the Conservatoire botanique national méditerranéen de Porquerolles and the Regione Liguria (Decree n. 859 of 22/08/2018).

## Data Availability

Raw AFLP data are available at Zenodo:

Gabriele Casazza, & Carmelo Macri. (2020). AFLP matrix *Lilium pomponium* [Data set]. Zenodo. DOI 10.5281/zenodo.3786974.

Population data are also available at Zenodo:

Gabriele Casazza, & Davide Dagnino. (2020). Phytosociological dataset [Data set]. Zenodo. DOI 10.5281/zenodo.3757743.

## Supplemental Information

Supplemental information for this article can be found online at http://dx.doi.org/10.7717/peerj.11039#supplemental-information.

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
