# Peer review of "When ecological marginality is not geographically peripheral: exploring genetic predictions of the centre-periphery hypothesis in the endemic plant Lilium pomponium"

_PeerJ, doi:10.7717/peerj.11039_

## Round 0.1 · original submission · Major Revisions

The paper presents an analysis of genetic data to test the centre-periphery hypothesis. The approach is technically sound and the data are valuable.

I have some concerns:

1) Potential confounding effects should be controlled for the variables.
2) There has been debate over which index should be used to estimate genetic diversity and genetic differentiation. It might be worth trying a few different measurements to validate the statement.
3) The manuscript will benefit from further English language improvement and editing.

Reviewer 1 ·

Basic reporting

The submission by Casazza et al. is for the most part written well and easy to follow. There are a few places where a series of sentences begin with phrases such as 'consequently' or 'therefore' that I found a bit distracting, but this is a stylistic issue. The scholarship here is first rate, with a detailed list of appropriate citations that support the main points that Casazza et al. seek to make. I found the structure of the manuscript to be appropriate, and figures and tables were easy to interpret.

Experimental design

The work here clearly fits into the scope of PeerJ. The design of Casazza et al is an effective exploration of the Center-periphery hypothesis. The CPH assumes that the 'optimal' environment is in the center of the geographic range, but this assumption should be tested. Casazza et al. design their study in an insightful manner by asking first if geographic and environmental centre-periphery gradients are correlated and second by asking how genetic differentiation and genetic diversity are associated with these gradients. Their study has a very good sample size. The genetic data (AFLPs) that they have collected are be limited for some research questions and applications but are perfectly appropriate here because they represent a quick survey of genomic diversity. The description of what they did was easy to interpret and follow, so I didn't have many questions about how or why the authors did what they did. I have one substantial concern that I'll describe in the next section.

Validity of the findings

Underlying data are provided, and this work is reproducible. My main critique concerns the climatic variables that they used. Climatic variables are potentially correlated with one another, which makes me question whether the authors should use all of the variables in the Chelsa database. I would suggest that rather than using all of these data, Casazza et al. should identify and remove variables that are autocorrelated. This will likely make the spatial gradients in these variables more obvious and detectable, which in turn would provide a better estimate of the degree to which the climatic and genetic variation as associated.

Additional comments

no additional comments

Reviewer 2 ·

Basic reporting

Clear and unambiguous language is used throughout. A few noted corrections and suggestions are noted in the comments to author. For the most part, the background, literature, etc. are sufficient. The formatting of the article is appropriate. I can't seem to find a statement indicating where the assembled datasets (climate, AFLP, etc.) are/will be available, so I have not viewed any raw data. Existing tables are appropriate but insufficient; I would add a table of accession information and/or a statement about population location. Were vouchers made, and if so, where were they deposited. If not, why? It is conceivable that this species may be the target of poaching. The results of the data analyses do address the questions stated at the end of the introduction.

Experimental design

This study is structured as an exploration 1) of the correlation between geographic and environmental gradients, 2) the genetic association between geo. and env. gradients and both genetic differentiation and genetic diversity. The research question is unfortunately poorly defined, mainly because of vagueness in the connection between the CPH in the introduction and the questions. Structuring the paper around subsidiary hypotheses might help guide the reader to a better understanding and interpretation of the results.

The study seems to have been conducted in a rigorous fashion and with appropriate ethical considerations (specifically, a scientific collection permit was obtained). I am concerned that the explanation of the AFLP protocol is misleading. The authors correctly note that the standard AFLP technique is inappropriate for Lilium because the large genome results in excessive fragments. The authors state that they use the method used by Sakazono et al. 2012 with Lilium longiflorum to reduce the number of fragments. Whereas conventional AFLP uses MSE and ECO primers with one additional base each in the preselective amplification and three additional bases in the second, selective amplification. This appears to be what was done in this study (lines 128-132). The Sakazono method differs in that they used THREE bases on the MSE primer for the preselective amplification and FIVE additional bases on it for selective amplification. This inconsistency needs to be resolved. I am not able to review the use of climatic variables.

Validity of the findings

It is not possible to fully gauge the validity of the results given the absence of the raw data and the concerns over the use of the AFLP technique. The statistical analysis seems appropriate for the genetic data. The patterns discovered in the data include an increase in genetic differentiation near the geographic periphery of the species range, and a reduction in genetic diversity in ecologically marginally habitats. The "refugia within refugia" hypothesis is consistent with the data for the former pattern, for the latter, the conclusions as stated are speculative. The data doesn't really allow for a measure of drift-based vs. selection-based explanations for the reduction in diversity in poorer habitats.

Additional comments

Line 39: "environmental" is misspelled.

Line 76 and onwards: This reviewer would appreciate a brief synopsis on studies that failed to find population parameters in line with the CPH hypothesis, and why genetic diversity does follow the predictions of the hypothesis. The following line, states that "Nevertheless, other studies suggest that the genetic diversity of populations is mainly affected by the environment." It is unclear how diversity can be affected by the environment if it is not mediated through the population parameters such as demography, population size, etc. that supposedly do not conform to predictions. Continuing in the introduction, the organization could be improved to more clearly connect the expected relationships of genetic differentiation and diversity with geographic and environmental gradients. It is unclear what findings are expected in this species. For instance, on line 94; the authors state that the "environment is therefore expected to have had a strong impact on... genetic variation": what would this look like? Increased heterozygosity? Deficiency in private alleles in populations? How would this differ from the effects of geographic isolation? A suggestion to clarify this would be to develop this part of the introduction explicitly in terms of the main forces of evolutionary change involved: selection, drift, mutation, migration, non-random mating.

Line 97: change "their" to "there".

Line 111: Table 1 is the AFLP primers, not a sample list. Table 2 lists population numbers but does not provide localities, so the statement that sampling covers the geographical range cannot be evaluated.

Line 116: The second sentence beginning on this line "The AFLP..." should be moved to a position after the description of DNA isolation- before "for the AFLP..." on Line 122.

Line 118: Muratovic (2010) is not in the lit cited.

Line 264: The statement that "genetically impoverished populations are more prone to genetic drift and consequently to divergence" is confusing cause and effect: both genetic impoverishment and divergence are expected consequences of genetic drift, as allele frequencies change stochasitcally in populations with small effective size. Note that this presents a problem for the idea that natural selection may be playing a role in reducing genetic variation (the "genetic draft" hypothesis discussed beginning on line 317) since populations subject to high levels of drift will be correspondingly unresponsive to selection.

Lines 290-300: the logic of this argument is contorted but I think it could be made clearer.

Line 311-313: was habitat quality specifically characterized? COULD it be characterized by working backward from the climatological analysis? (I don't know the answer to this.) If it could, then perhaps genetic diversity could be evaluated specifically against quality.

Line 329: species' (geographical/environmentalal) distribution

Line 332: PeerJ asks that speculation be identified as such. It seems that the relative contribution of drift-related effects and selection is not something the authors wish to take a position on, so the balance must remain speculative.

---

## Round 0.2 · Minor Revisions

The reviewer has identified that a few improvements need to be made such as language editing and the modification of the introduction section.

Reviewer 3 ·

Basic reporting

1.English is not clear and concise enough. It is suggested to modify the language and expression.

Experimental design

no comment

Validity of the findings

no comment

Additional comments

In the introduction, the author said that when plants encounter climate change, they will migrate to a new and more adaptive environment, and their traits will change. But the CPH theory is that plant diversity and differentiation will gradually decrease and increase with the geographical center to the geographical edge. I don't think that the variation of plant traits from the geographical center to the edge of CPH theory seems to have much to do with the adaptation of plants to the new environment and the variation of traits. It is suggested that the author should revise the introduction to meet the main idea of this study.

---

## Round 0.3 · accepted · Accept

I am satisfied with the current version of the manuscript.